# Six-Minute Walk Distance Is a Useful Outcome Measure to Detect Motor Decline in Treated Late-Onset Pompe Disease Patients

**DOI:** 10.3390/cells11030334

**Published:** 2022-01-20

**Authors:** Kristl G. Claeys, Ann D’Hondt, Lucas Fache, Koen Peers, Christophe E. Depuydt

**Affiliations:** 1Department of Neurology, University Hospitals Leuven, 3000 Leuven, Belgium; ann.dhondt@uzleuven.be (A.D.); lucas.fache@student.kuleuven.be (L.F.); 2Laboratory for Muscle Diseases and Neuropathies, Department of Neurosciences, KU Leuven, 3000 Leuven, Belgium; christophe.depuydt@kuleuven.be; 3Department of Physical Medicine and Rehabilitation, University Hospitals Leuven, 3000 Leuven, Belgium; koen.peers@uzleuven.be

**Keywords:** glycogen storage disease type 2, GSD2, LOPD, 6MWD, muscle strength, Biodex^®^ dynamometer, isometric, longitudinal, ERT, enzyme replacement therapy

## Abstract

Late-onset Pompe disease (LOPD) is a rare, progressive disorder characterized by limb–girdle muscle weakness and/or respiratory insufficiency, caused by acid alpha-glucosidase (*GAA*) gene mutations and treated with enzyme replacement therapy. We studied isometric muscle strength in eight muscle groups bilaterally using a Biodex^®^ dynamometer, as well as the Medical Research Council sum score (MRC-SS), hand grip strength, 6 min walk distance (6MWD), 10 m walk test (10MWT) and timed up-and-go test (TUG) in 12 adult, ambulatory, treated LOPD patients and 12 age-/gender-matched healthy controls, every 6 months for 2 years. The mean isometric muscle strength showed a significant decline in right and left knee extensors at 12 months in controls (*p* < 0.014; *p* < 0.016), at 18 months in patients (*p* < 0.010; *p* < 0.007) and controls (only right side, *p* < 0.030) and at 24 months in both groups (*p* < 0.035). The mean 6MWD in patients significantly decreased after 24 months, from 451.9 m to 368.1 m (*p* < 0.003), whereas in controls, the mean 6MWD significantly increased after 6 months (*p* < 0.045) and 18 months (*p* < 0.020) (at 24 months *p* = 0.054). In patients and controls, the MRC-SS, hand grip test, 10MWT and TUG did not show significant changes (*p* > 0.05). We conclude that the 6MWD is a useful outcome measure to detect motor decline in treated LOPD patients.

## 1. Introduction

Late-onset Pompe disease (LOPD; also known as glycogen storage disease type 2 or GSD2) is a rare autosomal recessive disorder caused by acid alpha-glucosidase (GAA) deficiency. The lack of the GAA lysosomal enzyme results in the accumulation of glycogen in muscle cells, leading to progressive limb–girdle muscle weakness and respiratory insufficiency [1,2]. The onset and severity of Pompe disease largely depends on the residual GAA enzyme activity: the disease either develops during the first months of life as the classic severe infantile-onset Pompe disease (IOPD) [3], or later in life with a milder phenotype known as late-onset Pompe disease (LOPD) [2]. Current treatment consists of enzyme replacement therapy (ERT) with recombinant human alglucosidase alfa [4], (non-)invasive ventilation and physiotherapy.

In previous studies showing the effect of ERT in LOPD patients, the 6 min walk distance test (6MWD) was mainly used as a consistent positive outcome measure of motor function in this disease [5,6,7,8,9,10,11,12,13,14,15,16,17,18,19,20,21,22,23,24]. In contrast, motor strength as an outcome measure using the manual Medical Research Council sum score (MRC-SS) showed inconsistent results, with a significant improvement in ERT-treated LOPD patients in some studies [17,21], but without amelioration in others [9,15]. More recently, the Biodex^®^ dynamometer has been introduced in the neuromuscular field to assess muscle strength in an objective, quantitative manner, particularly in patients with Duchenne muscular dystrophy [25], hereditary inclusion body myopathy [26] and in a small study with four treated LOPD patients [18].

In this study, we assessed isometric muscle strength measurements in eight muscle groups bilaterally using the Biodex^®^ dynamometer in adult, ambulatory, ERT-treated LOPD patients and in age-/gender-matched controls every 6 months for a duration of 2 years. We also evaluated the MRC-SS, hand grip strength, 6MWD, 10 m walk test (10MWT) and timed up-and-go test (TUG) as outcome measures in this patient group, and compared the data with age- and gender-matched healthy controls.

## 2. Patients and Methods

### 2.1. Patients and Controls

We included 12 adult Belgian patients with genetically confirmed and symptomatic (i.e., presence of muscle weakness) LOPD and 12 gender- and age-matched, healthy control individuals. All patients were ambulatory and treated with alglucosidase alfa 20 mg/kg intravenously (Myozyme^®^, Sanofi-Genzyme, Genzyme Corporation, Cambridge, MA, USA). The Ethical Committee Research of UZ/KU Leuven approved the study (S-60965; date of approval: 20 December 2017). We obtained written informed consent from all study participants.

### 2.2. Muscle Strength Assessment Using Biodex^®^ Dynamometer

We measured isometric muscle strength using a quantitative Biodex^®^ dynamometer (Biodex System 4, Procare Belgium and Biodex Medical Systems, Shirley, NY, USA) in all patients and controls every 6 months for a study duration of 2 years. All measurements were performed by the same investigator to avoid inter-investigator variability. The isometric muscle strength of knee flexors and extensors was measured in sitting position with the knee at 60°, hip flexors and extensors were assessed in supine position with the hip at 60°, elbow flexors and extensors with the elbow at 60° and shoulder abductors and adductors with the shoulder at 60°. Muscle groups were assessed bilaterally. The order of muscle strength testing was held constant with strength of the knee flexors/extensors assessed first, then elbow, shoulder and lastly, the hip muscles. Prior to the first session, participants were familiarized with the Biodex^®^ dynamometer to avoid confounding strength changes due to exercise training or greater familiarity with the test equipment. There were three five-second contractions performed consecutively by each muscle group with 10 s rests between contractions. The participant was verbally encouraged during the test to perform maximum contraction. The peak torque output for each muscle group (in Newton meters, Nm) was used in the analysis.

### 2.3. Additional Muscle Strength and Motor Function Assessments

In addition, we assessed muscle strength using the manual 80-point MRC-SS and hand grip strength (in kilograms) of the dominant hand (right hand in all participants) using a Jamar^®^ hand dynamometer (Jamar Technologies, Hatfield, PA, USA). We measured motor function using the 6MWD (in meter), 10MWT (in seconds) and TUG (in seconds).

### 2.4. Pulmonary Function Tests

In the patients, Forced Vital Capacity (FVC) was measured at study onset (visit 1) and at the end of the study (visit 5), both in sitting and supine position, following standard procedures. FVC was measured in liters (L) and in percent decrease (%) compared to controls matched for age and sex, height and body weight.

### 2.5. Statistical Analysis

We used MedCalc^®^ for statistical analyses (MedCalc Software, Ostend, Belgium) [27]. Descriptive statistics are stated as averages (minimum–maximum) and percentages. We applied paired t-tests for the comparison of outcome variables between baseline visit (V1) and visits at months 6 (V2), 12 (V3), 18 (V4) and 24 (V5). If assumptions for normality were not met, non-parametric equivalents (Wilcoxon signed-rank test or sign-test) were applied. An unpaired t-test was applied for the comparison between left and right in patients and control subjects. Analysis of variance (ANOVA) with Bonferroni correction for multiple comparisons was applied to analyze the differences between the means at the different visits (V1–V5) for each of the outcome measures in both LOPD patients and controls. Significance level was determined at α = 0.05.

## 3. Results

### 3.1. Demographics and Clinical Characteristics of LOPD Patients

In both patient and control groups (n = 12), we included five males (42%) and seven females (58%) (Table 1). The mean age at study entry was 51.3 years (range 22–67) and 50.9 years (range 23–64), respectively. The mean age at symptom onset in LOPD patients was 32.8 years (range 1–52). At the time of study inclusion, all patients were of adult age, ambulatory, symptomatic showing muscle weakness and treated with alglucosidase alfa. The mean disease duration was 18.4 years (range 0.5–36), and the mean duration of ERT therapy at the time of start of the study was 8.8 years (range 0.5–13). Only one patient (i.e., patient 7) had an ERT duration at the time of study inclusion of less than 2 years (i.e., 0.5 years), whereas all the other patients had ERT treatment durations of much longer than 2 years, i.e., between 6 and 13 years. The 6MWD in patient 7 also showed a slight deterioration during the study (data from visit 1 to visit 5: 594 m, 508 m, 486 m, 474 m and 477 m) and did not have an impact on the reported results, significances or conclusions.

At the time of the study, one third of the patients (4/12) were non-invasively ventilated during the night.

Statistical analyses in the patients (n = 12) did not show significant changes in FVC for the study duration of two years, both in the sitting and in supine position (*p* > 0.05). In one patient, supine FVC measurements were not possible due to respiratory insufficiency in the supine position (patient 3, Table 1). Therefore, since in our study group there was no progressive respiratory insufficiency for the duration of the study, the decline in the 6MWD cannot be explained by changes in respiratory function.

### 3.2. Results of Biodex^®^ Dynamometer and Other Outcome Measures in LOPD Patients

In LOPD patients, the mean isometric muscle strength measured using a Biodex^®^ dynamometer showed a significant deterioration in the knee extensors bilaterally at 18 months (right: *p* < 0.010; left *p* < 0.007) and 24 months (right: *p* < 0.002; left: *p* < 0.017) compared to baseline (Table 2; Figure 1A,B). At the baseline visit, the mean muscle strength at the knee extensors was 79.5 Nm ± 43.4 (right) and 79.3 Nm ± 42.6 (left), whereas after 18 months, the mean muscle strength significantly decreased to 67.7 Nm ± 41.3 (right) and 67.1 Nm ± 39.6 (left), and after 24 months, to 66.2 Nm ± 39.6 (right) and 65.1 Nm ± 39.0 (left) (Table 2). The mean isometric muscle strength in the hip flexors on the right side showed a significant change after 18 months (*p* < 0.031), but this effect did not sustain after 24 months and was not present at the left side. No significant decrease in the mean isometric muscle strength was measured for the study duration of 24 months in any of the other muscle groups (hip extensors, knee flexors, shoulder abductors and adductors and elbow extensors and flexors) using the Biodex^®^ dynamometer (*p* > 0.05). There were no significant differences between the mean isometric muscle strength in the different muscle groups measured at the left and right side in patients (*p* > 0.05).

In patients with LOPD, the mean 6MWD significantly decreased after 24 months (*p* < 0.003), from 451.9 m at baseline to 368.1 m after 2 years, corresponding to a mean decline of 83.8 m (Table 2; Figure 1C). We compared the 6MWD between the LOPD patients with non-invasive ventilation (n = 4) and those without non-invasive ventilation (n = 8), and there was no significant difference in the 6MWD between the two groups (*p* > 0.05) and no difference in the 6MWD trend for the study duration (Table 1; Figure 1D). All patients remained ambulatory during the study. The MRC-SS, hand grip test, 10MWT and TUG did not show a significant change during the study (*p* > 0.05; Table 2). There were no significant differences between the means at the different visits (V1–V5) for each of the outcome measures in LOPD patients (ANOVA, *p* > 0.05; Table 2).

### 3.3. Results of Biodex^®^ Dynamometer and Other Outcome Measures in Controls

Similarly to LOPD patients, in the age- and gender-matched control individuals, the mean isometric muscle strength measured using a Biodex^®^ dynamometer also showed a significant decline in the knee extensors at 12 months at both sides (right: *p* < 0.014; left: *p* < 0.016), at 18 months only at the right side (*p* < 0.030) and at 24 months at both sides (right: *p* < 0.007; left: *p* < 0.035) compared to baseline (Table 3; Figure 1A,B). At the baseline visit, the mean muscle strength at the knee extensors was 147.7 Nm ± 64.5 (right) and 133.0 Nm ± 56.7 (left), whereas after 12 months, the mean muscle strength significantly decreased to 125.9 Nm ± 56.5 (right) and 111.0 Nm ± 41.7 (left), and after 24 months to 112.2 Nm ± 40.2 (right) and 111.8 Nm ± 41.3 (left) (Table 3).

The mean isometric muscle strength in the knee flexors on the right side showed a significant decrease after 12 months (*p* < 0.034) and 24 months (*p* < 0.020), but this effect was not measured at 18 months and was not present at the left side (*p* > 0.05). No significant decrease in the mean isometric muscle strength was measured for the study duration of 24 months in any of the other muscle groups (hip extensors, hip flexors, shoulder abductors and adductors and elbow extensors and flexors) using the Biodex^®^ dynamometer (*p* > 0.05). In comparison to the patient group, there were no significant differences between the mean isometric muscle strength in the different muscle groups measured at the left and right side in controls (*p* > 0.05).

In contrast to LOPD patients, in controls, the mean 6MWD significantly increased after 6 and 18 months, from 661.3 m at baseline to 688.2 m after 6 months (*p* < 0.045), to 680.2 m after 18 months (*p* < 0.020), to 694.6 m after 24 months, which just failed to reach significance (*p* = 0.054), corresponding to a mean amelioration of 33.3 m over 2 years (Table 3; Figure 1C). This increase in the 6MWD in control individuals is probably due to a training effect. Similarly to the patient group, the MRC-SS, hand grip test, 10MWT and TUG did not show a significant change during the study (Table 3; *p* > 0.05). There were no significant differences between the means at the different visits (V1–V5) for each of the outcome measures in controls (ANOVA, *p* > 0.05; Table 3).

## 4. Discussion

Our study showed that the 6MWD is a useful outcome measure to detect motor decline in treated LOPD patients. In contrast, quantitative isometric strength measurement using a Biodex^®^ dynamometer, MRC-SS, hand grip strength, 10MWT and TUG proved not to be suitable outcome measure in this group of patients for a study duration of 2 years.

The 6 min walk distance (6MWD) was originally developed in 2002 as an integrated assessment of pulmonary, cardiac, circulatory and muscular capacity in patients with moderate to severe lung disease and provides a measure of the functional exercise level needed to perform daily physical activities [28]. Since then, several studies have also used the 6MWD in neuromuscular diseases as an endpoint to assess muscular function during disease progression or treatment, such as in Duchenne muscular dystrophy [29,30], hereditary inclusion body myopathy [26], spinal muscular atrophy [31] or metabolic myopathies including LOPD [8,11,12,15,24,32].

In the LOPD patients in our study, the mean baseline 6MWD was 452 m ± 143, whereas other studies reported lower baseline 6MWDs from 246 to 376 m [8,9,11,12,15,32]. These lower values can be explained by differences in age, disease duration or treatment duration at study inclusion. For comparison, the mean baseline 6MWD in our healthy age- and gender-matched controls aged 23–64 years (mean 51 years) was 661 m ± 82, similar to 571 m ± 90 in another study in healthy adult controls aged 40–80 years (median 58 years) that also showed significantly shorter distances in controls above 60 years [33].

After 2 years of treatment, 6MWD significantly declined in LOPD patients with a mean distance of 84 m. A deterioration of 6MWD in treated LOPD patients has been shown in other studies as well after 2–3 years of treatment following an initial improvement [15,21,23,34]. In contrast to these studies, the baseline in our study did not correspond to the start of treatment but represented a mean treatment duration of 8.8 years. Therefore, the mean decline in 6MWD in our study cannot be compared directly with the other reports. Moreover, since the 6MWD at the time point of start of treatment in our LOPD patient group is not known, the 6MWD at 2-year follow up might still be higher than the initial value at start of treatment, similarly to the findings in other studies [15,21,23,34]. In untreated LOPD patients, the 6MWD has been shown to be lower than in treated patients [11]. In our study, we did not include a group of untreated LOPD patients, since this comparison was not the objective of the study, and not treating LOPD patients when a treatment is available would not be acceptable from an ethical point of view.

A change in the mean distance in 6MWD in patients of 83.8 m (representing about 18% of the initial mean value of 451.9 m) represents a clinically meaningful change according to literature data [35]. In controls, a mean change of 33.3 m (which ranged from 3.5 to 32 m at 18 months) over an initial value of 661.3 m represents a 5.0% change (in line with reported SEM% for 6MWD), which does not reach the clinically meaningful change in the 6MWD as established by the literature and as expected in controls. Since the MDC value in meters is expected to change in relation to the condition of patients versus controls, our data in controls suggest that a variation in the 6MWD higher than 5% may be considered, even in patients, as the minimal detectable change and it is likely to reflect a true change rather than a measurement error, while variations < 5% may be measurement errors/training effects and so on. Indeed, the patients show a change in the 6MWD much higher than 5%, even when other motor measures are stable, reflecting the sensibility of this measure and its ability to detect little changes.

The 6MWD is easy to perform, quick and inexpensive, but can only be used in ambulatory patients and depends on motivation, age, sex, height, weight and skeletal problems, which can influence gait and thereby affect the distance walked. However, the 6MWD will usually not be the only endpoint in clinical trials, and in LOPD patients, parameters for respiratory function such as forced vital capacity (FVC) will also be included, as well as patient-reported outcome measures (PROMs), which are becoming more and more important, such as the Rasch-built Pompe-specific Activity (R-Pact) scale, measuring daily-life activities, with a proven positive correlation with physical outcomes and developed specifically for LOPD [36,37].

In contrast to the 6MWD, which we would recommend using as an endpoint in clinical trials in LOPD based on our own data and others, we showed that isometric strength measurement using a Biodex^®^ dynamometer was not a suitable outcome measure in LOPD patients for the study duration of two years. However, we cannot exclude the idea that the muscle strength measurement using Biodex^®^ might be a good outcome measure when a longer observation period would be considered. In most of the tested muscle groups, we did not find a significant change over the study duration of 24 months. Only in the knee extensors was there a significant consistent and symmetrical decline in muscle strength measured over 2 years. In comparison to the literature data, the knee extensors in LOPD patients are better preserved with longer disease duration compared to, e.g., the hip extensors or knee flexors, which are already affected early in the disease course [38]. This can also be seen in our results in LOPD patients in Table 2: at the start of the study, the highest muscle strength can be measured in the knee extensors compared to all other tested muscle groups. The fact that muscle strength in the knee extensors is clearly higher to start with at the onset of the study might explain why, especially in those muscles, a significant decline can still be detected over the duration of the study. However, not only in the patients but also in the controls, a decrease in muscle strength in the knee extensors was measured, which might be related to aging and/or other error sources such as the lack of motivation, the selection of controls, a biased examiner since the assessments were of course not blinded, etc.; this did not influence the functional capacity in the 6MWD, which increased in the controls over the 2 years of study duration. The decreased muscle strength in patients might be explained by disease progression, but the factor of aging might also partially contribute to the decline in isometric strength measurement. Thus, since the muscle strength of the knee extensors using the Biodex^®^ dynamometer not only decreased in patients but also significantly decreased in healthy controls even though their 6MWD increased during the study duration of 2 years, we concluded that these Biodex^®^ measurements are not applicable as a reliable and functionally relevant outcome measure in clinical trials in LOPD patients.

A few other studies in neuromuscular disorders have measured muscle strength using a Biodex^®^ dynamometer; however, most of them were cross-sectional [25,26], in contrast to our longitudinal study design. One study concerning only four treated LOPD patients performed strength measurements using Biodex^®^ and concluded there was a small increase in muscle strength after 2–6 years follow up [18].

We showed that in our study group of LOPD patients, there was no progressive respiratory insufficiency for the duration of the study, both in the sitting and in the supine position (*p* > 0.05). Therefore, the decline in the 6MWD cannot be explained by changes in respiratory function in our patient group. However, considering that the parameter that deteriorates is the 6MWD and not muscle strength, it is conceivable that the aspect involved might be functional endurance, corresponding to other recent studies [39]. However, it might also still be possible that the 6MWT is more capable of detecting small declines in distinct functions, taken together (endurance, strength, posture, respiration), that perhaps single outcome measures (FVC alone, and so on) cannot catch.

Finally, if the TUG, 10MWT, MRC-SS, dynamometry (for most muscles) did not change over a 2-year period but only the 6MWD declined, it might also be argued that the disease is quite stable over the years with ERT, after a mean ERT treatment duration of 8.8 years (range 0.5–13 years), with the exception of distance walked on the 6MWD. It is noteworthy that the MRC score has an intrinsically low reliability due to substantial inter-rater and intra-rater variability. Furthermore, the TUG and 10MWT are also timed tests like 6MWD, but in these short-timed measures, there is a certain degree of error and less reliability due to the short duration of the tests. A correlation with a disease-specific patient-reported outcome measure (PROM), such as the Rasch-built Pompe-specific Activity (R-Pact) scale, might also have helped to identify a real decline from the patient’s perspective [36,37].

We conclude that the 6MWD is a useful outcome measure to detect motor changes in treated ambulatory late-onset Pompe disease patients and should be included as an endpoint in clinical trials in LOPD. Further studies are needed to also analyze the proper outcome measure for non-ambulatory LOPD patients.

## Figures and Tables

**Figure 1 cells-11-00334-f001:**
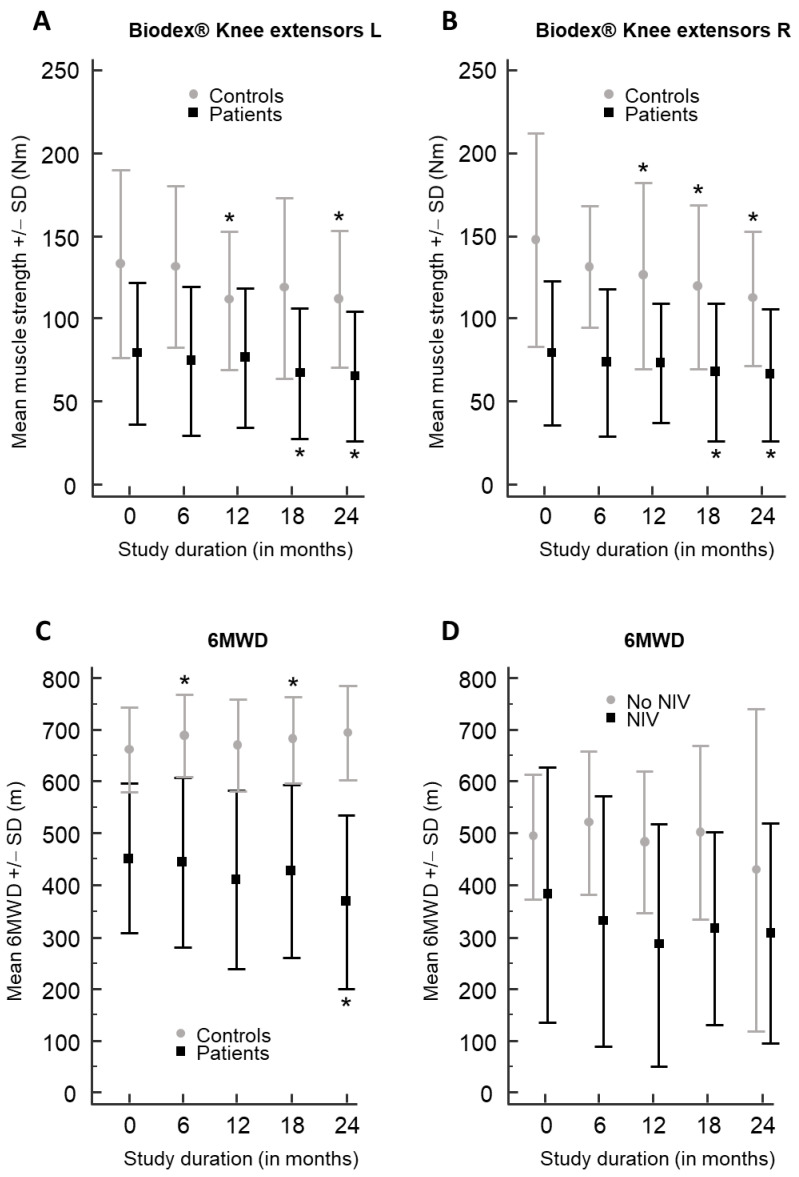
Isometric muscle strength results using Biodex^®^ in knee extensors and 6MWD in patients and controls. Results of the mean isometric muscle strength (±1 standard deviation, SD) in Newton meters (Nm) of the left (**A**) and right (**B**) knee extensors measured using the Biodex^®^ dynamometer are shown for LOPD patients (black) and controls (gray) for the study duration of 24 months. In (**C**), the mean 6MWD (±1 standard deviation, SD) in meters (m) is presented for patients (black) and control individuals (gray) for the study duration of 24 months. In (**D**), the mean 6MWD (±1 standard deviation, SD) in meters (m) is presented for patients with non-invasive ventilation (NIV) (black) and those without NIV (gray). In panels (**A**–**C**), an asterisk (*) indicates a significant difference with the baseline value (0 months).

**Table 1 cells-11-00334-t001:** Clinical and genetic features of the LOPD patients included in the study.

ID	Gender	Age at Symptom Onset (y)	Symptoms at Onset	*GAA* Mutations	Age at Study Inclusion (y)	Disease Duration at Inclusion (y)	Duration of ERT (y)	NIV at Night (Y/N), Age at Start NIV (y)	FVC Sitting Visit 1 (0 Months) (%)	FVC Supine Visit 1 (0 Months) (%)	FVC Sitting Visit 5 (24 Months) (%)	FVC Supine Visit 5 (24 Months) (%)
1	F	42	LW	c.-32-13T>G; c.482_483delCC	48	6	6	N	101	97	96	94
2	F	1	LW	c.-32-13T>G; c.923A>C	22	21	13	N	106	91	106	107
3	M	39	R	c.-32-13T>G; c.258dupC	61	22	11	Y (40)	33	ND	30	ND
4	F	35	LW	c.-32-13T>G; c.1548G>A	49	14	12	Y (36)	48	33	51	38
5	M	17	LW	c.-32-13T>G; c.482_483del	43	26	8	N	73	67	76	62
6	F	27	hyperCK, F	c.-32-13T>G; c.525delT	40	13	8	N	114	93	121	105
7	M	42	LW	c.-32-13T>G; c.2608C>T	54	12	0.5	Y (53)	76	45	84	45
8	M	45	LW	c.-32-13T>G; c.1681_1699dup19	67	22	11	Y (60)	101	49	105	59
9	F	44	LW	c.-32-13T>G; del exon 18	62	18	10	N	70	43	64	32
10	F	52	LW	c.-32-13T>G; c.258dupC	63	11	8	N	108	78	73	56
11	F	25	LW	c.-32-13T>G; c.186dup11	61	36	9	N	84	64	79	62
12	M	25	LW	c.-32-13T>G; c.1075G>A	45	20	9	N	80	62	79	56
		32.8 (1–52)			51.3 (22–67)	18.4 (0.5–36)	8.8 (0.5–13)					

ID, patient number; y, years; *GAA*, acid alpha-glucosidase gene; ERT, enzyme replacement therapy; NIV, non-invasive ventilation; FVC, forced vital capacity; F, female; M, male; LW, limb–girdle weakness; R, respiratory weakness; hyperCK, increased creatine kinase in blood; F, fatigue; Y, yes; N, No; ND, not done.

**Table 2 cells-11-00334-t002:** Results of Biodex^®^ dynamometer and other outcome measures in LOPD patients.

Outcome	Visit 1 (0 Months)		Visit 2 (6 Months)					Visit 3 (12 Months)					Visit 4 (18 Months)					Visit 5 (24 Months)					ANOVA
Measures	Mean	±SD	Mean	±SD	V2 vs. V1	95%CI	*p*	Mean	±SD	V3 vs. V1	95%CI	*p*	Mean	±SD	V4 vs. V1	95%CI	*p*	Mean	±SD	V5 vs. V1	95%CI	*p*	*p*
Hip ext. R	54.1	19.8	62.0	29.6	7.7	−0.4 to 15.8	0.061	59.6	24.7	6.0	−3.6 to 15.7	0.191	59.4	27.1	5.3	−2.5 to 13.2	0.162	61.5	28.5	7.4	−2.5 to 17.3	0.126	0.959
Hip ext. L	55.5	26.4	62.3	31.1	6.7	−2.1 to 15.5	0.119	59.1	30.3	5.6	−4.8 to 16.0	0.253	60.1	28.4	4.6	−4.3 to 13.5	0.274	57.8	24.3	2.3	−9.5 to 14.1	0.677	0.986
Hip flex. R	19.2	14.8	22.5	14.2	3.7	−1.8 to 9.2	0.162	23.9	16.6	4.2	−1.9 to 10.4	0.156	24.9	11.5	5.8	0.6 to 10.9	**0.031**	26.4	15.6	7.3	−1.0 to 15.5	0.079	0.813
Hip flex. L	17.9	12.4	20.9	13.8	3.3	−1.5 to 8.2	0.153	22.0	15.4	3.5	−2.7 to 9.7	0.233	23.1	14.2	5.2	−1.6 to 11.9	0.119	22.1	12.8	4.2	−2.1 to 10.5	0.170	0.921
Knee ext. R	79.5	43.4	73.5	44.5	−3.3	−8.5 to 2.0	0.194	73.2	35.9	−6.4	−18.1 to 5.3	0.256	67.7	41.3	−11.8	−20.2 to −3.5	**0.010**	66.2	39.6	−13.3	−20.5 to −6.2	**0.002**	0.937
Knee ext. L	79.3	42.6	74.8	44.8	−2.4	−7.7 to 2.9	0.332	76.7	42.0	−2.6	−13.3 to 8.2	0.609	67.1	39.6	−12.1	−20.3 to −4.0	**0.007**	65.1	39.0	−14.1	−25.2 to −3.1	**0.017**	0.901
Knee flex. R	52.6	24.7	51.9	25.7	−1.3	−4.6 to 1.9	0.378	44.3	21.6	−8.2	−16.6 to 0.2	0.055	48.1	24.6	−4.4	−9.7 to 0.9	0.095	47.6	23.9	−5.0	−11.0 to 1.0	0.093	0.919
Knee flex. L	47.4	21.3	48.2	24.8	0.4	−3.2 to 4.0	0.807	42.3	20.4	−5.1	−11.0 to 0.7	0.081	46.6	22.0	−0.9	−4.2 to 2.5	0.577	44.9	22.2	−2.5	−6.2 to 1.2	0.167	0.970
Shoulder abd. R	18.8	5.0	18.6	6.1	0.4	−3.4 to 4.2	0.821	18.3	8.7	0.4	−4.2 to 5.1	0.839	17.7	6.7	−0.4	−2.9 to 2.1	0.726	16.8	6.1	−1.2	−3.4 to 0.9	0.223	0.953
Shoulder abd. L	16.8	6.8	16.9	7.8	1.1	−2.4 to 4.5	0.498	17.5	6.9	0.8	−1.5 to 3.2	0.437	16.2	7.0	−0.7	−3.6 to 2.2	0.596	17.3	7.1	0.6	−2.1 to 3.2	0.646	0.993
Shoulder add. R	42.3	14.4	40.6	14.5	−1.2	−3.5 to 1.2	0.291	40.1	14.8	−2.1	−6.2 to 1.9	0.269	41.2	15.0	−0.6	−3.9 to 2.7	0.696	41.6	15.3	−0.3	−3.4 to 2.8	0.853	0.997
Shoulder add. L	47.3	14.9	44.7	13.9	−2.2	−6.0 to 1.7	0.236	46.5	13.7	−1.0	−5.2 to 3.3	0.627	45.5	14.0	−1.7	−6.3 to 3.0	0.443	47.3	14.5	−0.1	−5.1 to 4.8	0.952	0.991
Elbow ext. R	40.8	13.8	40.9	13.8	1.8	−4.4 to 8.0	0.534	42.1	14.9	1.2	−4.0 to 6.4	0.614	43.3	14.3	2.5	−3.4 to 8.4	0.378	40.0	13.0	−0.9	−7.0 to 5.3	0.765	0.982
Elbow ext. L	44.8	16.9	45.7	13.1	3.2	−1.5 to 8.0	0.159	43.2	16.3	−1.6	−6.0 to 2.8	0.446	44.8	15.5	0.0	−4.7 to 4.7	0.988	43.1	13.7	−1.7	−7.0 to 3.7	0.506	0.993
Elbow flex. R	22.6	10.9	21.0	10.2	−1.4	−6.0 to 3.1	0.506	22.0	12.2	−0.6	−6.0 to 4.8	0.810	22.2	11.2	−0.5	−4.7 to 3.7	0.805	22.7	11.7	0.0	−5.2 to 5.2	0.995	0.997
Elbow flex. L	22.2	14.1	22.3	12.3	2.1	−1.1 to 5.4	0.178	22.7	12.1	0.5	−4.0 to 4.9	0.816	22.9	12.0	0.7	−2.3 to 3.7	0.626	23.1	11.2	0.9	−3.2 to 4.9	0.646	1.000
Hand grip R(kg)	37.3	10.7	39.3	8.9	1.1	−1.3 to 3.5	0.323	37.4	7.9	−0.8	−5.1 to 3.5	0.680	37.5	9.3	−0.5	−4.0 to 3.0	0.743	36.8	7.7	−2.2	−7.1 to 2.7	0.328	0.974
MRC-SS (/80)	67.2	8.2	70.4	7.7	1.0	−5.5 to 7.5	0.733	71.1	8.3	1.8	−4.9 to 8.5	0.556	70.4	8.3	1.2	−5.8 to 8.1	0.709	71.3	8.1	2.3	−5.3 to 9.8	0.506	0.785
6MWD (m)	451.9	143.3	443.9	163.6	−19.1	−58.8 to 20.6	0.305	410.6	172.2	−41.3	−86.9 to 4.3	0.071	427.6	167.0	−28.5	−63.9 to 6.9	0.102	368.1	167.6	−60.6	−92.0 to −29.1	**0.003**	0.826
10MWT (s)	8.4	2.8	8.6	2.8	0.2	−0.7 to 1.2	0.626	9.4	3.7	1.0	−0.3 to 2.4	0.127	9.0	3.8	0.7	−0.4 to 1.7	0.214	9.1	3.5	0.8	−1.3 to 2.9	0.414	0.960
TUG (s)	7.4	4.7	7.8	5.8	0.2	−1.0 to 1.3	0.779	8.2	5.4	0.6	−0.5 to 1.7	0.265	7.8	5.4	1.1	−0.3 to 2.5	0.112	9.0	6.2	1.3	−1.4 to 4.0	0.289	0.981

Biodex^®^ measurements in Newton meters (Nm); SD, standard deviation; V, visit; V1, baseline visit (0 months); 95%CI, 95% confidence interval; *p*, *p*-value; ANOVA, analysis of variance; L, left; R, right; ext., extensors; flex., flexors; abd., abductors; add., adductors; hand grip, hand grip test (in kilograms); MRC-SS, Medical Research Council sum score (on a maximum of 80 points); 6MWD, 6 min walk distance (in meter); 10MWT, 10 m walk test (in seconds); TUG, timed up-and-go test (in seconds). Significant values are underlined and indicated in bold.

**Table 3 cells-11-00334-t003:** Results of Biodex^®^ dynamometer and other outcome measures in controls.

Outcome	Visit 1 (0 Months)		Visit 2 (6 Months)					Visit 3 (12 Months)					Visit 4 (18 Months)					Visit 5 (24 Months)					ANOVA
Measures	Mean	±SD	Mean	±SD	V2 vs. V1	95%CI	*p*	Mean	±SD	V3 vs. V1	95%CI	*p*	Mean	±SD	V4 vs. V1	95%CI	*p*	Mean	±SD	V5 vs. V1	95%CI	*p*	*p*
Hip ext. R	107.7	31.8	115.6	41.2	3.1	−14.7 to 20.9	0.705	110.4	47.5	2.7	−24.4 to 29.7	0.833	108.3	42.8	−2.1	−23.4 to 19.3	0.833	109.5	36.3	−7.1	−25.9 to 11.6	0.407	0.992
Hip ext. L	100.1	33.3	115.4	38.1	12.1	−8.9 to 33.0	0.229	102.3	35.1	2.2	−21.3 to 25.6	0.843	105.9	39.9	3.6	−21.3 to 28.5	0.755	107.2	32.1	−1.8	−24.6 to 21.1	0.864	0.872
Hip flex. R	52.1	22.6	51.7	19.3	−3.9	−10.4 to 2.6	0.207	45.5	16.0	−6.6	−15.9 to 2.6	0.142	42.7	19.5	−8.8	−19.6 to 2.1	0.102	46.0	17.8	−5.2	−12.6 to 2.2	0.141	0.723
Hip flex. L	47.1	22.5	48.8	22.6	−1.2	−5.6 to 3.3	0.570	45.1	21.7	−2.1	−10.8 to 6.7	0.610	46.6	24.6	−0.8	−13.8 to 12.3	0.896	42.7	19.8	−5.6	−20.7 to 9.6	0.422	0.980
Knee ext. R	147.7	64.5	131.4	36.4	−23.1	−49.7 to 3.5	0.082	125.9	56.5	−21.8	−38.3 to −5.3	**0.014**	119.1	49.4	−29.7	−55.8 to −3.6	**0.030**	112.2	40.2	−32.4	−53.2 to −11.7	**0.007**	0.557
Knee ext. L	133.0	56.7	131.7	48.8	−8.1	−24.2 to 8.0	0.289	111.0	41.7	−22.0	−39.0 to −5.0	**0.016**	118.4	54.5	−16.4	−35.5 to 2.8	0.086	111.8	41.3	−21.6	−41.3 to −2.0	**0.035**	0.725
Knee flex. R	98.4	34.9	100.8	40.4	−1.1	−11.1 to 9.0	0.819	89.2	35.3	−9.2	−17.5 to −0.8	**0.034**	92.2	43.7	−4.1	−15.4 to 7.1	0.431	86.1	27.4	−9.5	−17.1 to −2.0	**0.020**	0.880
Knee flex. L	94.4	39.1	98.0	45.0	−1.5	−11.8 to 8.8	0.754	86.7	38.5	−7.7	−16.9 to 1.6	0.095	89.6	42.7	−4.0	−15.9 to 7.9	0.474	86.6	28.8	−8.9	−22.0 to 4.1	0.153	0.951
Shoulder abd. R	32.4	21.1	32.0	21.2	−0.4	−3.9 to 3.2	0.827	29.6	21.4	−2.8	−7.0 to 1.4	0.167	31.0	22.7	−1.1	−5.5 to 3.3	0.593	27.3	12.5	−1.8	−8.3 to 4.7	0.535	0.981
Shoulder abd. L	29.8	16.8	30.1	16.7	0.3	−2.6 to 3.1	0.843	28.2	18.0	−1.6	−5.3 to 2.0	0.345	27.8	16.7	−1.3	−4.1 to 1.4	0.297	24.9	12.7	−1.5	−4.4 to 1.4	0.265	0.958
Shoulder add. R	78.2	37.5	74.2	31.6	−4.0	−12.6 to 4.6	0.333	70.6	25.5	−7.6	−18.0 to 2.9	0.140	73.2	31.9	−4.3	−16.2 to 7.7	0.444	67.9	24.5	−7.7	−22.3 to 6.8	0.256	0.954
Shoulder add. L	73.3	31.9	69.9	27.8	−3.4	−8.6 to 1.8	0.180	69.3	22.6	−4.0	−13.0 to 5.0	0.348	69.5	28.3	−3.3	−13.7 to 7.0	0.490	64.1	18.6	−5.4	−21.1 to 10.2	0.446	0.959
Elbow ext. R	44.6	19.4	44.3	19.6	−0.3	−3.6 to 3.1	0.864	43.8	17.2	−0.7	−4.7 to 3.2	0.693	43.6	18.8	0.2	−3.5 to 4.0	0.892	40.3	13.0	−2.4	−10.2 to 5.4	0.500	0.986
Elbow ext. L	46.3	19.2	45.5	21.1	−0.8	−3.1 to 1.5	0.470	44.9	13.9	−1.4	−6.3 to 3.6	0.552	44.5	18.2	−0.5	−5.8 to 4.9	0.853	43.5	14.5	0.2	−5.1 to 5.5	0.937	0.998
Elbow flex. R	37.2	20.8	36.2	20.1	−1.0	−2.7 to 0.8	0.249	36.0	17.0	−1.1	−4.5 to 2.2	0.470	32.8	18.3	−4.1	−11.3 to 3.1	0.230	32.8	16.2	−2.1	−6.6 to 2.3	0.298	0.971
Elbow flex. L	32.4	18.6	34.6	18.2	2.2	−1.0 to 5.4	0.159	31.8	15.2	−0.6	−4.6 to 3.4	0.742	31.0	16.5	−0.9	−4.9 to 3.1	0.623	29.3	11.4	0.7	−4.3 to 5.7	0.752	0.964
Hand grip R(kg)	36.5	9.6	34.9	9.5	−1.6	−4.5 to 1.2	0.233	34.7	12.2	−1.8	−5.8 to 2.1	0.331	36.2	12.7	−0.3	−3.5 to 2.8	0.814	35.4	11.5	−0.9	−3.2 to 1.4	0.401	0.993
MRC−SS (/80)	80.0	0.0	80.0	0.0	0.0	0.0 to 0.0	1.000	80.0	0.0	0.0	0.0 to 0.0	1.000	80.0	0.0	0.0	0.0 to 0.0	1.000	80.0	0.0	0.0	0.0 to 0.0	1.000	1.000
6MWD (m)	661.3	81.8	688.2	78.7	16.0	0.5 to 31.5	**0.045**	670.4	88.8	9.2	−9.2 to 27.5	0.295	680.2	82.9	17.7	3.5 to 32.0	**0.020**	694.6	91.0	23.3	−0.4 to 47.1	0.054	0.896
10MWT (s)	4.2	1.4	4.7	1.4	0.5	−0.4 to 1.3	0.242	4.4	1.2	0.2	−0.5 to 0.9	0.486	4.5	1.1	0.3	−0.6 to 1.1	0.502	4.2	0.9	0.2	−0.7 to 1.1	0.702	0.867
TUG (s)	1.8	0.3	1.8	0.5	0.0	−0.2 to 0.2	0.927	1.9	0.9	0.2	−0.4 to 0.7	0.540	2.1	1.0	0.2	−0.3 to 0.8	0.357	1.8	0.5	0.0	−0.3 to 0.3	1.000	0.799

Biodex^®^ measurements in Newton meters (Nm); SD, standard deviation; V, visit; V1, baseline visit (0 months); 95%CI, 95% confidence interval; *p*, *p*-value; ANOVA, analysis of variance; L, left; R, right; ext., extensors; flex., flexors; abd., abductors; add., adductors; hand grip, hand grip test (in kilograms); MRC-SS, Medical Research Council sum score (on a maximum of 80 points); 6MWD, 6 min walk distance (in meter); 10MWT, 10 m walk test (in seconds); TUG, timed up-and-go test (in seconds). Significant values are underlined and indicated in bold.

## Data Availability

Data supporting reported results can be obtained from the corresponding author on reasonable request.

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
