# Peer review of "Six-Minute Walk Distance Is a Useful Outcome Measure to Detect Motor Decline in Treated Late-Onset Pompe Disease Patients"

_cells, 2022, doi:10.3390/cells11030334_

Round 1
Reviewer 1 Report
The authors have now addressed all my comments and suggestions satisfactorily.
Author Response
The reviewer mentioned that all his/her comments and suggestions were addressed satisfactorily.
Reviewer 2 Report
I think that the paper "Six-minute walk distance but not isometric strength measure- 2 ment is a useful outcome measure to detect motor decline in 3 treated late-onset Pompe disease patients" provides useful data with respect to the proposed outcome measure to monitor the course of Pompe disease . The data are presented in a clear and complete way and the suggested insights have been adequately addressed in the text
Author Response
The reviewer mentioned that the suggested insights have been adequately addressed in the text.
Reviewer 3 Report
see attached file

Reviewer 4 Report
The study is well designed and the manuscript is well structured.
The aim of the longitudinal study by Kristl Claeys et al. is to evaluate different outcome measures for the assessment of muscle strength in late onset Pompe disease (LOPD) patients on enzyme replacement therapy (ERT) and in normal age and gender matched controls. Isometric muscle strength measurement using Biodex dynamometer in 4 antagonistic proximal muscle groups in upper and 4 antagonistic proximal muscle groups in lower limbs, Medical Research Council sum score, hand grip strength, 6-minute walk distance (6MWD), 10-meter walk test and up-and-go test were implemented. The authors concluded that only 6MWD is a useful outcome measure to detect motor changes in treated LOPD patients. The authors concluded that other tests implemented, including isometric muscle strength measurement using Biodex dynamometer, are not suitable.
The manuscript is clear, relevant for the field of assessment of muscle strength in LOPD patients on ERT and is well structured. References are appropriate.
The manuscript is scientifically sound: The study is well designed and the methods used are appropriate. Healthy controls were chosen for comparison not LOPD patients without treatment due to ethical reasons since not treating LOPD patients when treatment is available is not acceptable from the ethical point of view.
The details given in the “Patients and methods” allow the reproducibility of the procedures.
Figures and tables properly demonstrate the data and are easy to interpret and understand. Statistical analysis using paired t-test for comparison of outcome variables between baseline visit and visits at 6, 12, 18 and 24 months when normality of the distribution was met is appropriate (non-parametric statistics were used if normality was not met). Analysis of variance with Bonferroni correction for multiple comparisons to analyse the differences between the means at different visits for each outcome measure is also appropriate.
Please modify your conclusion regarding “suitability of Biodex dynamometer”; line 88 page 13 and in the abstract line 24, 25. You have demonstrated that Biodex dynamometer detected deterioration in knee extensors at 18 and 24 months, so for knee extensors “Biodex” is appropriate; it seems that it does not detect the deterioration of more severely affected muscles, as flexors in general.
Please modify also the title in accordance with the remark above.
